# Extracellular Vesicles as Carriers of Adipokines and Their Role in Obesity

**DOI:** 10.3390/biomedicines11020422

**Published:** 2023-02-01

**Authors:** Tamara Camino, Nerea Lago-Baameiro, María Pardo

**Affiliations:** 1Grupo Obesidómica, Área de Endocrinología, Instituto de Investigación Sanitaria de Santiago de Compostela (IDIS), Xerencia de Xestión Integrada de Santiago (XXIS/SERGAS), 15706 Santiago de Compostela, Spain; 2Centro de Investigación Biomédica en Red de la Fisiopatología Obesidad y Nutrición (CIBERobn), Instituto de Salud Carlos III, 28029 Madrid, Spain

**Keywords:** extracellular vesicles, exosomes, adipose tissue, adipokines, obesity

## Abstract

Extracellular vesicles (EVs) have lately arisen as new metabolic players in energy homeostasis participating in intercellular communication at the local and distant levels. These nanosized lipid bilayer spheres, carrying bioactive molecular cargo, have somehow changed the paradigm of biomedical research not only as a non-classic cell secretion mechanism, but as a rich source of biomarkers and as useful drug-delivery vehicles. Although the research about the role of EVs on metabolism and its deregulation on obesity and associated pathologies lagged slightly behind other diseases, the knowledge about their function under normal and pathological homeostasis is rapidly increasing. In this review, we are focusing on the current research regarding adipose tissue shed extracellular vesicles including their characterization, size profile, and molecular cargo content comprising miRNAs and membrane and intra-vesicular proteins. Finally, we will focus on the functional aspects attributed to vesicles secreted not only by adipocytes, but also by other cells comprising adipose tissue, describing the evidence to date on the deleterious effects of extracellular vesicles released by obese adipose tissue both locally and at the distant level by interacting with other peripheral organs and even at the central level.

## 1. Introduction

Obesity has reached pandemic proportions worldwide and continues to increase at an alarming rate; specifically, the World Health Organisation (WHO) estimates that 39% of adults over the age of 18 are overweight or obese [1]. Thus, there is an urgent need to gain better knowledge of the molecular mechanisms involved in the deregulation of energy metabolism, as well as more in-depth knowledge about new signals secreted by adipose tissue in both healthy physiological and pathological conditions, as energy imbalance causes ectopic accumulation and inflammation of fat, and hyperplasia and hypertrophy of adipocytes throughout the body. Adipose tissue is known to be an endocrine organ capable of communicating both locally and peripherally with other tissues, and even centrally, by interacting with appetite control areas in the hypothalamus through secreted molecules called adipokines that are involved in the immune responses and the metabolic regulation through their paracrine and/or endocrine actions [1]. Obesity manifests itself when there is an imbalance between energy intake and expenditure, and, consequently, there is an accumulation of energy excess in the form of triglycerides in the adipocytes causing hypertrophy and hyperplasia, which leads to an increase in the mass of adipose tissue that becomes inflamed and fibrotic [1]. Altogether, this leads to the altered secretion of adipokines, participating in obesity-associated comorbidities, such as insulin resistance, inflammation, hypertension, cardiovascular diseases, type 2 diabetes, non-alcoholic fatty liver disease (NAFLD), metabolic disorders, and cancer, and, overall, shortening life expectancy [2]. Above all, it is important to highlight that the coronavirus (COVID-19) pandemic has brought to light a higher risk of severe outcomes for those patients with obesity and obesity-related complications [3]. In particular, it has been observed that COVID-19 patients show an altered glycoprotein profile, including a decrease in serum adiponectin and a hyperglycaemic state presenting a state of insulin resistance [4]. However, it is still unclear how adipokines and cytokines are involved in obesity-related diseases and how these signaling molecules vary depending on the excess location of different adipose tissue depots. In this context, different alternative communication pathways have recently emerged for adipose tissue, such as non-classical protein secretion [5], the release of extracellular vesicles [6], or the release of long non-coding RNAs [7] and microRNAs [8] that can travel freely or within extracellular vesicles.

Extracellular vesicles (EVs) are small, rounded membrane spheres that can range in size from 30–1000 nm. Depending on their biogenesis and size, EVs can be microvesicles (100 nm–1 µm) derived from plasma membrane blebbing, exosomes (30–100 nm) assembled into multivesicular endosomes (MVE) that are secreted by exocytosis, or large vesicles (50–5000 nm) that include apoptotic bodies released by cells preceding apoptosis [9,10]. Interestingly, EVs are released by all cells in the body [11], so they can be detected both in the extracellular environment and in virtually all body fluids, such as blood, urine, cerebrospinal fluid, saliva, and even tears, etc. Recently, exosomes, microvesicles, and other EVs have emerged as potentially good biomarkers of disease with potential use for monitoring pathology or even the efficacy of therapeutic treatments; this is because EVs contain membrane and cytosolic components, such as RNA, proteins, and lipids, and this composition is conditioned by the site of biogenesis [12]. Most interestingly, due to this bioactive cargo, EVs can exert stimulatory or inhibitory functional effects, such as cell proliferation, apoptosis, cytokine production, immune modulation, or metastasis, by inducing cellular transduction signals or genetic and epigenetic changes [13,14] and, thus, modifying the characteristics and activity of specific target cells and tissues, mediating intercellular and interorgan crosstalk [13,14].

In adipose depots, adipocyte derived EVs, together with those liberated by endothelial, immune, and mesenchymal stem cells, play an important role in adipogenesis, targeting tissue formation, modulation of the immune microenvironment, adipokine release, as well as tissue remodeling [15]. Furthermore, EVs released by adipose tissue exert their metabolic actions in distal organs (such as the liver, pancreas, skeletal muscle, and brain) by sending genetic information to specific target cells to regulate gene expression. In particular, EVs released from brown adipose tissue have recently attracted considerable interest. This tissue type is the main site of adaptive thermogenesis, and there is evidence associating its activity with protection against obesity and metabolic diseases because of its ability to utilize glucose and lipids for thermogenesis [16]. Consequently, active brown adipose tissue is present in healthy adult humans and compromised in patients with obesity. It has been proven that brown depots exert systemic beneficial effects through their secretory function [17]. Thus, in vitro studies have shown that brown adipocytes, and even the intermediate known beige ones, secrete EVs [18,19]. Therefore, increasing evidence suggests that EVs secreted by adipose tissues play an important role in the regulation of metabolic inflammation, energy metabolism, and insulin sensitivity [15].

## 2. Extracellular Vesicles Secreted by Adipose Tissue in Obesity

Currently, the great challenge surrounding EVs research, apart from deciphering the exact functional role and characterizing their dynamic content, is to find a feasible, accurate, and reliable method to isolate these particles in a reproducible manner, allowing to obtain pure EVs, while also maintaining their functional characteristics [20], thus, allowing the use of these EVs in clinical applications [21]. Among the various established techniques for EV isolation are differential centrifugation, density gradient centrifugation, immunoaffinity, size exclusion chromatography (SEC), and polymer-based precipitation protocols [22]; however, all of them have drawbacks, such as the isolation of unwanted contaminants or being tedious procedures [23]. Therefore, with the intention to minimize this issue, the International Society for Extracellular Vesicles (ISEV) developed a consensus guideline comprising the minimum experimental requirements for the study of EVs (MISEV) [20]. Once the vesicles have been isolated, it is important to evaluate and confirm the method used to obtain these particles; in this regard, the ISEV recommends the use of different complementary vesicle characterization techniques, e.g., quantitatively defining the source of EVs (for example, number of secretory cells, volume of biological fluid, tissue mass); determining the number and size of particles using NTA (nanoparticle tracking analysis), TEM (transmission electron microscopy), cytometry, ZetaView and/or the ExoView platform; characterizing the vesicle content to determine the presence of components associated with the various EVs subtypes and, thus, rule out co-isolated unwanted components produced by the isolation method [20].

In the context of obesity, several studies have characterized EVs isolated from primary cell cultures of adipocytes or adipose tissue stem cells by various techniques, ranging from in vitro cell cultures of mesenchymal cells differentiated into adipocytes, to those secreted by whole adipose tissue at different anatomical locations and under different metabolic status. The study by Connolly et al. characterized the EVs secreted during adipocyte differentiation (day 0 to 15 of differentiation) in the 3T3-L1 murine cell line, observing a decrease in the amount of particle secretion measured by NTA from day 0 to 15 of adipocyte differentiation without showing changes in particle size [24]. This result agrees with our own work, in which we characterized the EVs secreted by adipocytes of the murine line C3H10T1/2, before and after differentiation, and also after being exposed to different treatments to promote insulin resistance and lipid hypertrophy [25]. Specifically, in this study, we showed that healthy control adipocytes secrete a higher concentration of EVs than pathological adipocytes (hypertrophied and insulin-resistant) [25].

On the other hand, the study by Akbar et al. isolated EVs from primary cultures of human adipocytes extracted from adipose tissue explants (0.5 g) from the gluteal and abdominal area of healthy donors by ultracentrifugation and filtration [26]. In this research, they observed that adipocytes from the gluteal fat secreted around 6.10 × 10^7^ particles/mL and measured approximately 115 nm. In contrast, adipocytes from the abdominal fat secreted around 7.60 × 10^7^ particles/mL with an average size of 125 nm. Notably, both types of EVs were positive for endosomal markers such as TSG101 (tumor susceptibility gene 101) [26]. In another study, the number and size of large (lEVs) and small (sEVs) EVs secreted by Adipose-Derived Stem Cells (ADSCs) isolated from subcutaneous tissue of healthy human donors were characterized by various techniques such as TEM, NTA, flow cytometry and immunodetection [27]. The lEVS had a mean size of 388 nm, while the sEVs 138 nm. In particular, the sEVs were positive for vesicle markers such as tetraspanins CD81 and CD63, syntenin-1, and flotillin-1 [27]. In this respect, a study by Franquesa et al. observed that ADSCs from subcutaneous adipose tissue secreted approximately 2.5 × 108 particles/mL [28].

In contrast to previous studies, our group has characterized EVs released from human and murine whole adipose tissue considering the color type (white vs. brown), anatomical location (subcutaneous vs. visceral), and metabolic status (obese vs. lean) [29,30]. The advantage of this whole tissue approach is that it maintains the 3D tissue structure, preserving intercellular communication, including inflammatory signals and cell–Extracellular Matrix (ECM) interaction, which is characteristically affected in obesity [31]. By this means, it was observed that human obese visceral adipose tissue secretes a greater number of EVs than subcutaneous [29]. This result agrees with previous proteome studies of our group, where it was also observed that visceral adipose tissue secretomes contained a greater number of proteins, cytokines, and other metabolites than subcutaneous adipose tissue [32]. Of note, we have observed that protein concentration, from the same initial cellular/tissue amount, is higher in EVs isolated from pathological adipocytes and from obese visceral adipose tissue than in vesicles from healthy control adipocytes or from obese subcutaneous, which may indicate differences in the amount of protein load for each metabolic situation [25,29]. Moreover, those EVs isolated from whole adipose tissue explants had an average size from approximately 50–80 nm or from approximately 75–175 nm, depending on the characterization technique, ExoView or NTA platform, respectively. These EVs were positive for different vesicle markers such as CD81, CD9, CD63, Alix, syntenin-1, and/or lysosome-associated membrane glycoprotein 1 (LAMP1) [25,29].

Of interest is the work of Chen and collaborators, in which they demonstrate that murine brown adipocytes also secrete exosomes whose production augments when simulating cold exposure and β-adrenergic stimulation induced by cAMP treatment [33]; this effect was also observed when whole mouse adipose tissue was exposed to cold. Likewise, they show that beige, but not white, adipocyte exosome secretion increases more than 10-fold after treatment with cAMP.

In summary, studies on the characterization of the size, nature, and concentration of EVs released from adipose tissue shows different profiles depending on the site of origin and on the metabolic statutes (obese and lean), finding a higher concentration of EVs in secretomes isolated from pathological adipocytes or visceral adipose tissue from patients with obesity compared with healthy adipocytes. This suggests that the populations and kinetics of secreted vesicles are highly dynamic being also modulated by the physiology of the cell of origin. Therefore, we can affirm that adipose tissue secretes vesicles of various sizes and different nature, and that their concentration depends on the type of tissue and nutritional status.

### 2.1. Adipose Tissue Extracelular Vesicles Cargo Content

EVs have been postulated as key factors in various physiological processes, such as development, tissue homeostasis, aging, metabolic regulation, circadian rhythms, lactation, and in pathological processes, such as in various non-infectious diseases (cancer, inflammation, metabolic, immune, or respiratory disorders) and infectious diseases (malaria, Changas disease, sleeping sickness, among others) [34,35]. In this context, several studies on the biological role of EVs in various human pathologies have emerged providing a new vision and a paradigm shift in biomedical research [35,36]. Specifically, in metabolic diseases, several studies suggest that EVs are key players in the communication of metabolic organs in physiological homeostasis at both paracrine and endocrine levels, likely with an important contribution in the metabolic disturbance related to obesity, diabetes, and associated comorbidities [37]. Thus, research work in this field is opening new lines of investigation focused on the development of new therapeutic strategies for obesity and related diseases through EVs released by the involved endocrine organs [16,37]. However, the level of development and understanding of EVs and their role in metabolic diseases is still incipient compared with the progress in other pathological processes, such as cancer [38]. In particular, there are very interesting reports showing the involvement of vesicles secreted by white adipose tissue in insulin signaling in the liver and in muscle cells [39]. Moreover, these studies show how EVs participate in a reciprocal proinflammatory loop between adipocytes and macrophages that aggravates local and systemic insulin resistance (IR) [40,41], induces alterations of the transforming growth factor beta pathway in hepatocytes [42], attracts macrophages, regulates appetite and body weight at the central level [43], or even participating in obesity-related cancer [44]. Interestingly, exosome secretion by brown adipose tissue has also been described, especially after BAT activation [34].

The functionality of EVs is mainly determined by their cargo, whereby their membrane antigens and molecular cargo are crucial to exert a functional effect on a target cell/tissue. Thus, functional role of EVs is related to their ability to interact with target cells and release their content into these cells, which gives EVs an important part in cellular communication and targeted delivery as therapeutic vehicles [36]. In addition to this implication in cell communication, EVs have emerged as carriers of new diagnostic and prognostic pathway biomarkers for the detection and monitoring of diseases for various relevant reasons; for instance, EVs are very dynamic and reflect the functional state of the cell of origin, they can be found in all body fluids, and they allow a minimally invasive analysis, without the need of biopsies permitting the sequential collection of samples [45]. Under this premise, it is vital to know the composition (proteins, lipids, RNAs) of EVs to elucidate changes or alterations associated with the physiology of the cell of origin [46]. Unlike other tissues, the composition of EVs released from adipose tissue under normal and pathological conditions has been poorly studied. However, there are important advances in this field, and different publications have described several non-coding RNAs as exosomal microRNAs from adipose tissue, many of them secreted by MSCs (mesenchymal stem cells), related to endocrine and paracrine metabolic regulation [47,48] (Figure 1).

#### 2.1.1. Adipose Tissue Extracellular MicroRNAs

MicroRNAs (miRNAs) are a group of short (20–24 nucleotides) single-stranded non-coding RNAs with the ability to regulate gene expression at the post-transcriptional level when bound to the mRNA of a target gene [43]. miRNAs, similar to EVs, are also found in all biological fluids, such as blood, urine, and extracellular fluids, and can also be packaged within lipid or lipoprotein complexes, such as microvesicles, exosomes, or apoptotic bodies. Specific miRNAs are preferentially secreted by cells and are, therefore, more likely to be found at the circulatory level [49]. Precisely, different miRNAs have been shown to play a key role in adipose biology as they are involved in the regulation of white, beige [50], and brown adipose tissues’ differentiation and function [49] (Figure 1 and Table 1).

MiRNAs bind to the RNA-induced silencing complex (RISC) and, subsequently, through RISC, they bind to the 3’ untranslated regions (3’ UTR) of the mRNAs leading to translation repression or mRNA degradation [43]. Initially, RNA polymerase acts by transcribing the miRNAs as polyadenylated 5′-capped precursors known as primary miRNA (pri-miRNA) [49]. This pri-miRNA is then matured by Drosha, which releases into the nucleus of the pre-miRNA, which is a small hairpin RNA. In turn, the pre-miRNA is exported to the cytoplasm by exportin-5 and then cleaved by DICER to give rise to a small RNA duplex. Two species of mature miRNAs can be generated from the 3′ end (passenger strand) and 5′ end (leader strand) of a pre-miRNA precursor, and although there is increasing coexistence of -5p and -3p miRNA species, in most cases, only one species remains viable, while the complementary species is degraded [49]. So far, miRNAs have been shown to regulate many biological processes and to be involved in the metabolism and pathophysiology of multiple diseases (angiogenesis, cancer, diabetes, and hepatic steatosis), while they can also serve as relevant diagnostic biomarkers [49]. There is growing evidence that fat is an important source of circulating miRNAs, and that miRNAs secreted by adipocytes, especially those within extracellular vesicles or exosomes, may assist in tissue-to-tissue communication acting as novel adipose hormones [50,51].

At the paracrine level, a study by Zhang et al. showed an enhanced population of 45 miRNAs in exosomes secreted from whole rat adipose tissue compared to vesicles that were secreted by adipose tissue-derived stem cells [52]. Of these, 14 miRNAs were involved in the regulation of adipogenesis, with miR-450a-5p being one of the most abundant and able to stimulate adipogenesis with the inhibition of WNT1-inducible-signaling pathway protein 2 (WISP2), a negative regulator of adipogenesis.

Furthermore, under the premise that EVs and their cargo molecules, in this case the miRNAs, play different roles at many levels of intracellular and interorgan crosstalk [53]. Research work by Hong Gao et al. demonstrates a novel role of adipose tissue macrophages (ATMs) in obesity-induced β-cell adaptation through miRNA release in extracellular vesicles [54]. By doing both in vivo and in vitro experiments, they demonstrated that ATM-EVs shed from obese mice inhibit insulin secretion and enhance β-cell proliferation; similarly, they describe this same effect in human islets after treatment with obese ATM EVs [54]. Notably, depletion of miRNAs reduces the effect of obese ATM EVs, as evidenced by the minimal effects of obese DicerKO ATM EVs on β-cell responses. Thus, it is concluded that ATM-derived EVs act as endocrine vehicles responsible for transporting miRNAs, and, subsequently, mediating obesity-associated β-cell adaptation and dysfunction [54]. In addition, others have shown how adipose tissue macrophages modify systemic metabolism via exosomal microRNAs [55]. Ying and collaborators reported how miR-155 within ATM-EVs were able to induce glucose intolerance and insulin resistance after their injection into lean mice. On the contrary, miR-155 KO mice improved their response to insulin and glucose uptake compared to controls. This result may suggest that ATM-derived exosomes are transporting crucial regulators of glucose homeostasis and insulin sensitivity to metabolic target cells of the liver and muscle. Therefore, miR-155 was found to be overexpressed in obese ATM exosomes and proved to suppress insulin effect on glucose production by downregulating targets such as peroxisome proliferator-activated receptor gamma (PPARγ) or glucose transporter type 4 (GLUT4) [55].

**Table 1 biomedicines-11-00422-t001:** Overview of the reviewed publications related to the characterization of the miRNA load in EVs shed by adipose tissue; miRNA type, function, tissue origin, target, species.

miRNA	Function(s)	Tissue Origin	Target(s)	Species
miR-450a-5p [52]	Positive regulator of adipogenesis	Adipose tissue	WISP2	Rat, in vitro
miRNA-155 [55]	Inductor of glucose intolerance and insulin resistance, impairs insulin secretion and enhances β cell proliferation	Adipose tissue macrophages exosomes	PPARγ, GLUT4	Mouse, in vitro
miR-99b [48]	Negative regulator of FGF21 expression	Brown adipose tissue	FGF21	Human, mouse, in vitro
miR-200a [56]	Capable of causing cardiomyocyte hypertrophy through downregulation of TSC1 and subsequent induction of mTOR signaling	Adipose tissue	TSC1	Mouse, in vitro
miRNA-23a/b [57]	Regulates the metabolism in tumorigenesis	Adipose tissue	VHL/HIF	Human, mouse, in vitro
miRNA-132-3p [58]	Endocrine factor regulating hepatic lipogenesis for cold adaptation	Brown adipose tissue	Srebf1	Mouse, in vitro

A recent study by Thomou et al. shows that humans with lipodystrophy, as well as a fat-specific knockout of the miRNA processing enzyme Dicer (ADicerKO), have decreased levels of circulating exosomal miRNAs [48]. They have identified 419 exosomal miRNAs with significantly decreased levels in ADicerKO mice compared to wild-type. Interestingly, they demonstrate that transplantation of adipose tissue (brown, inguinal, or epidydimal white fat) from the wild-type is able to restore miRNA levels, thus, reaffirming that adipose tissue is a key source of circulating exosomal miRNAs. On the other hand, Thomou et al. also found in ADicerKO mice a threefold increased level of circulating fibroblast growth factor 21 (FGF21), as well as a significant increase in the level of Fgf21 mRNA in fat, liver, muscle, and pancreas. In this case, transplantation of normal brown fat into ADicerKO mice was able to decrease the level of Fgf21 mRNA in the liver, which was also reflected with the decrease in FGF21 at the circulating level [48]. With this study, miR-99b was identified as a signal sent to the liver for the regulation of FGF21 expression. Furthermore, miR-99b was present in brown fat exosomes capable of binding to the 3′UTR of Fgf21 mRNA, thereby reducing FGF21 expression. Hence, these results suggest that miRNAs released by fat depots participate in the regulation of gene expression in the liver and cause an effect on whole organism metabolism [48].

Moreover, it was suggested that adipose tissue secretion of miR-200a loaded in exosomes could be the molecular mechanism underlying the adverse cardiovascular effect of rosiglitazone; thus, establishing a clear crosstalk between adipocytes and cardiomyocytes in mice [56]. In this report, they show how rosiglitazone, a PPARγ agonist used to treat diabetes as an insulin sensitizer, activates the expression and secretion of this miRNA by adipose tissue, causing cardiomyocyte hypertrophy through the diminution of Hamartin or Tuberous sclerosis 1 protein (TSC1) and subsequent induction of mTOR signaling [56].

Recently, at a different level, Yang Liu et al. have discovered how exosomal miRNAs are expressed in patients with hepatocellular carcinoma (HCC) and an elevated body fat ratio [57]. In this study, they found that microRNA-23a/b was significantly up-regulated in both serum exosomes and tumor tissues of those HCC patients with a high body fat ratio compared to those with low. Moreover, by performing in vitro assays, they provided evidence that this miRNA originates from adipocytes and is then transported by EVs to cancer cells, thus, promoting HCC cell growth and migration [57]. Taking the former into account, it can be suggested that adipose tissue implication in tumor progression may be mediated through miRNAs traveling in adipose tissue EVs.

Exosomal microRNAs derived from brown adipose tissue were also shown to be capable of influencing gene expression, specifically in the liver, as demonstrated by Kariba et al. by identifying exosomal miR-132-3p as an endocrine factor regulating hepatic lipogenesis for cold adaptation [58]. Furthermore, it was shown that norepinephrine, as a sympathetic nervous system neurotransmitter mediating cold-induced BAT activation, alters the composition of brown adipocyte-derived exosomal miRNAs (BAC); this is the case of miR-132-3p, which is significantly induced by this neurotransmitter. Thus, in this report, they found by in vitro assays, that isolated BAC exosomes suppressed the expression of hepatic sterol regulatory element-binding protein 1 (Srebf1), a predicted target of miR-132-3p. On the contrary, this effect was not observed in miR-132-3p-inhibited BAC [58].

Thus, increasing findings are evidence that adipose tissue miRNAs can cause effects by regulating gene expression in different diseases by means of their transport in extracellular vesicles. Precisely, further research regarding the role of miRNAs traveling within EVs in obesity will allow the identification of therapeutic targets for the prevention and treatment of obesity and its comorbidities.

#### 2.1.2. Adipose Tissue EVs Proteome

As in the case of vesicular miRNAs, there is also great interest in obtaining a better understanding of adipose tissue-specific EVs protein content, and, more precisely, in relation to their anatomical location since there is a clear role of body fat distribution in the metabolic complications of obesity; thus, visceral adipose tissue being considered more deleterious [59]. In addition, there is a need to better understand the specific vesicles secreted by the different cellular components of adipose tissue, including the immune cells that invade this tissue during the development of obesity. Therefore, there is an interest in studying how these vesicles enable cross-cellular communication involving inflammatory signals and the interaction of those cells that comprise adipose tissue with the surrounding extracellular matrix (ECM), which is characteristically altered in obesity [31]. Hence, a better understanding of the antigens and intracellular proteins that travel in EVs released from adipose tissue under physiological and pathological conditions may provide relevant information about this alternative cellular communication pathway. Furthermore, EVs from adipose tissue may provide a source of non-invasive biomarkers and reveal new therapeutic pathways [60] (Figure 1 and Table 2).

Kranendonk et al. published one of the earliest works describing the protein content of EVs isolated from human adipocytes of Simpson–Golabi–Behmel Syndrome (SGBS), which is a complex congenital overgrowth syndrome [41]. These vesicles were shown to carry FABP4 (fatty acid binding protein 4, adipocyte), adiponectin, TNF-α (tumor necrosis factor-alpha), MCSF (macrophage colony-stimulating factor), RBP-4 (retinol-binding protein 4), and MIF (macrophage migration inhibitory factor) [41]. On the other hand, Rosina et al. described EVs isolated from the primary culture of brown adipocytes extracted from the BAT of mice subjected to thermogenic stress, observing that these EVs contained mitochondrial proteins, such as PDHE1-β (pyruvate dehydrogenase beta, mitochondrial) [61]. Moreover, Eirin et al. performed a comparative proteomic analysis of extracellular vesicles isolated from porcine adipose tissue-derived mesenchymal stem cells in which a large variety of identified proteins were related to angiogenesis, blood coagulation, apoptosis, extracellular matrix remodeling, and regulation of inflammation [62].

In a research work focused on the relationship of obesity with cancer, Lazar et al. performed a proteome analysis of the EVs released by the murine adipocyte cell line 3T3-F442A differentiated in vitro in which they identified 324 proteins of which the majority were related to lipid metabolism [44]. In addition, the study by Durcin et al. showed a detailed analysis of the proteome of small (sEVs) and large (lEVs) vesicles released from a murine adipocyte cell line (3T3-L1) [63]. Notably, large EVs were enriched in membrane components, organelles, and cellular parts compared with small EVs, which were enriched in extracellular matrix (ECM) components [63]. At the pathological level, the study by Lee et al. showed differences in the exosomal protein profile as a result of the pathophysiological state of adipose tissue, specifically in vesicles released from adipocytes from primary cell cultures of obese diabetic and non-diabetic rats [64]. They show that EVs from the adipose tissue of rats with insulin resistance, obesity, hypertension, hyperglycemia, and hyperinsulinemia contain a higher ratio of aquaporin 7, caveolin, and lipoprotein lipase compared to vesicles from control animals [64].

On the other hand, studies of whole adipose tissue explants were performed, such as that of Kranendonk et al., in which they observed characteristic adipose tissue proteins, such as adiponectin and FABP4, and changes in the proteomic composition of EVs isolated from human adipose tissue explants from thin donors according to their anatomical location, subcutaneous and omental, with a higher number of inflammation-related proteins in EVs from the omental depot [41]. In addition, another interesting study analyzed the exosomal proteomic profile of adipose tissue in the context of maternal-fetal communication [65]. In this report, they show a quantitative proteomic analysis of EVs isolated from the omental adipose tissue secretome of pregnant women with normal glucose tolerance (NGT) compared with those with gestational diabetes mellitus (GDM). Their results evidenced the differential expression of proteins targeting the sirtuin signaling pathway, oxidative phosphorylation, and the mechanistic target of rapamycin signaling pathway in EVs from patients with GDM relative to EVs from women with NGT [65]. Moreover, recent studies have also identified novel adipokines through the proteomic profiling of small EVs isolated from rat inguinal adipose tissue [66]. In this work, they describe three new vesicular adipokines, nucleoplasmin-3, DAD-1, and STEAP3 metalloreductase, whose expression is altered in obese animals compared with lean animals (Sprague–Dawley rats) [66].

In recent years, our group has performed an extensive qualitative and quantitative proteome analysis of EVs shed by murine pathological adipocyte cell models that included lipid hypertrophy (palmitate and oleic acid) and insulin resistant adipocytes, by whole explants of subcutaneous and visceral adipose tissues from obese patients, and from whole explants of white and brown tissues of obese and lean rats [18,25,29,30]. Interestingly, this significant proteome analysis has shown many common proteins as those described in the previous studies mentioned above [30]. However, we identified many other proteins, such as leptin, that have not been previously described to the best our knowledge in adipose tissue vesicles. Remarkably, differences in the protein composition of EVs were observed according to the metabolic status of the cell/tissue of origin and the anatomical location of subcutaneous, visceral, and brown adipose tissues. Thus, these differences at the proteomic level were reflecting the metabolic insult, the cellular origin, and the nutritional status of the individual, suggesting the usefulness of these vesicular proteins as biomarkers of pathology [18,25,29]. Hence, in previous studies by our group, we described the first reference proteomic maps of EVs released by human obese adipose tissue giving their anatomical location, which shed some light on the requested need to characterize the protein load of EVs in the context of obesity [37]. Thus, we show that vesicles released by obese adipose tissue share structural vesicular proteins, but also contain depot- and metabolic status-specific components.

In our previous work, common proteins identified in EVs from all adipose tissues or obese adipocytes included structural and cytoskeletal proteins such as annexins A1/A6, which modulate anti-inflammatory processes that may be dysregulated in obesity-associated DM2 (type 2 diabetes) [67], as well as histones such as H4, whose epigenetic changes (lysine acetylation) are known to trigger metabolic disorders in mice with diet-induced obesity (DIO) [68]. Furthermore, interestingly, other known proteins related to fatty acid transport or synthesis, or adiposity, were also identified in all the analyzed obese EVs, such as perilipins (perilipin-1), caveolin-1 (cav-1), fatty acid-binding protein of adipocytes (FABP4), or fatty acid synthase (FAS) [41]. In particular, cav-1 and perilipin-1 proteins have already been proposed in the literature as vesicular biomarkers of adipose deposition [69], especially perilipin-1, which has been identified as a biomarker of adipose tissue at the circulating level [70,71]. Furthermore, perilipin-1 protein is one of the most abundant adipocyte proteins, which plays a key role in the regulation of lipid and glucose metabolism, being also associated with body weight and the related complications of obesity [72]. Thus, in our studies, we found that perilipin-1 protein is detected at a higher ratio in EVs released from fat depots under obese conditions, especially from visceral and brown fat, compared to EVs from lean fat [29]. Since EVs carrying caveolin-1 are known to be involved in communication between different cell types within adipose tissue, such as endothelial cells and adipocytes [69], and considering recent studies showing that this protein is influencing tumor development or progression by controlling metabolism through glycolysis, fatty acid metabolism, or mitochondrial pathways [73], the role of caveolin-1 traveling within EVs deserves further analysis. On the other hand, FABP4 protein was detected as elevated in EVs released from brown and visceral adipose tissue under obese conditions compared to EVs released from obese subcutaneous fat depots and under normal-weight conditions [29]. This finding is coherent with the circulating levels of this protein, which positively correlate with the incidence of metabolic disease, which may suggest that adipose tissue vesicular-FABP4 may be contributing to those levels detected in peripheral blood [74,75]. Another protein found in EVs secreted by adipose tissue in obese individuals was the FAS protein [29], whose high levels have previously been associated with impaired insulin sensitivity and adipose tissue dysfunction in obesity [76]. In the same context, the enrichment of transmembrane proteins and antigens in the vesicles released by whole adipose tissue explants in obesity is also significant, and, from these, the release of vesicles by other components of adipose tissue, such as mesenchymal or immune cells, can be suggested. Thus, we have shown that human vesicles from obese adipose tissue contain extracellular matrix (ECM) constituents (collagens, thrombospondin 1), ECM modifiers (MMP-1, 2, 9, 14, matrix remodeling-associated protein 5, TIMP-1), and ECM receptors (integrin β 2, CD44, CD36) [25,29] previously associated with obesity and insulin resistance [77]. Moreover, EVs released from subcutaneous fat depots of obese individuals were characterized by ECM proteins and proteins of interest, such as GDF-15, which is considered an anti-obesity agent [78].

Furthermore, we have observed that EVs secreted by visceral fat depots of patients with obesity showed a greater variety of proteins compared to subcutaneous, and more interestingly, the functional classification of these proteins revealed that many were previously related to adipose tissue, obesity, and to the obesity-associated inflammation. This is the case for leptin [40], IL-6 (interleukin-6) [40], CXCL5 (chemokine 5 with C-X-C motif) [79], GRP78 (endoplasmic reticulum chaperone BiP) [80], septin 11 [81], DDP-4 (dipeptidyl-peptidase 4) [82], syntaxin-8 [83], PAI-1 (plasminogen activator inhibitor) [84], or HSL (hormone-sensitive lipase) [85], among others, identified exclusively in vesicles released by obese visceral adipose tissue or by pathological adipocytes in culture (insulin resistant and lipid hypertrophied) [25,29]. Consequently, the identification of new adipokines in visceral adipose tissue vesicles from patients with obesity, and in EVs released from adipocytes with lipid hypertrophy, such as DDP4, mimecan, and ceruloplasmin, is also of interest. The presence of the transmembrane glycoprotein DDP4 in obese visceral vesicles is significantly elevated compared to subcutaneous. In addition to its role as an incretin GLP-1 (glucagon-like peptide-1) and GIP (gastric inhibitory polypeptide) inhibitor, this protein has recently been identified as an adipokine released to a greater extent by visceral adipose tissue, and, in particular, in obese and insulin-resistant patients, and has, therefore, been suggested as a biomarker of visceral obesity, insulin resistance, and metabolic syndrome [86]. Furthermore, this protein (DDP4) was previously identified in EVs released from differentiated human subcutaneous depot adipocytes in vitro [87]. Similarly, we have shown that visceral adipose tissue from patients with obesity, and adipocytes with lipid hypertrophy, secrete an increased number of EVs positive for the mimecan protein [25,29]. This protein, which is abundantly expressed in adipose tissue, is considered an adipokine involved in the control of satiety because it inhibits food intake by inducing IL-1 and IL-6 expression in the hypothalamus, independently of leptin signaling [88]. Therefore, we hypothesize that, as with leptin, this hormone (mimecan) at the vesicular level may be released in an increased form as compensation for the development of obesity. In the same context, the protein ceruloplasmin has also been found to be up-regulated in EVs from fat depots under conditions of lipid hypertrophy and obesity, in particular, in visceral and brown adipose tissues [18,29]. Furthermore, recent studies have observed a significant increase at circulating levels of this protein in obese individuals compared to lean, thus, postulating this protein as a biomarker of adiposity [89]. Therefore, we believe that ceruloplasmin is secreted in EVs at an increased rate during the development of obesity. Interestingly, one of the most remarkable findings of our work was the identification of vesicular TFGBI (transforming growth factor-induced ig-h3 protein) associated in our analysis with visceral fat deposition in obese conditions, and almost exclusively present in EVs released from adipocytes with insulin resistance (IR) [25,29]. Notably, the gene encoding this protein, *TGFBI*, has been described as a diabetes risk gene in both mice and humans [90]. Furthermore, we were able to detect that this protein in EVs at the circulating level was present at a higher ratio in those vesicles from the plasma of obese patients with diabetes compared to vesicles from donors without this pathology [29]. Therefore, we hypothesized that the EV-associated TGFBI protein might be a good candidate biomarker of early insulin resistance and progression of this pathology.

In addition, we also observed that the AHNAK protein, CD14, and vimentin are in a higher ratio in EVs from visceral fat depots of morbidly obese patients compared to subcutaneous fat vesicles [29]. The AHNAK protein plays a crucial role in body fat accumulation by regulating adipose tissue development and being involved in metabolic homeostasis [91]. In turn, the CD14 protein, also elevated in brown adipose tissue vesicles from obese compared to normal-weight animals [18], has previously been linked to macrophage infiltration in adipose tissue leading to an increased inflammatory state of this tissue [92]. On the other hand, the protein vimentin plays a key role in adipose tissue plasticity and has also been found to be involved in obesity and type 2 diabetes [93].

In contrast, decreased syntenin-1 protein was observed in EVs shed by human obese visceral adipose tissue compared to subcutaneous EVs in our study [29]. Additionally, other structural and/or small vesicle biogenesis associated proteins (CD316, Alix, CD98, lactadherin, among others) were detected in a higher ratio in human obese subcutaneous EVs compared to those present in EVs from obese visceral tissue [63]. It is also interesting to note the presence of IL-8 in subcutaneous vesicles from obese patients, as the presence of this cytokine is positively correlated with obesity-associated parameters, such as BMI (Body Mass Index), WC (waist circumference), and the C-reactive protein level, which indicates inflammation status [94]. These findings can be explained by the fact that EVs vary according to the location and type of adipose tissue; hence, EVs are shaped by physiological and pathological changes in the cell or tissue of origin. Thus, the visceral adipose tissue of obese individuals has been reported to be more inflamed than the subcutaneous [95], therefore, are invaded by macrophages and other immune cells, which also release EVs that can modulate the type, quantity, and dynamic release of these particles. Therefore, we postulate that the proteins comprised in obese visceral adipose tissue vesicles better represent the metabolic changes associated with obesity and its comorbidities that travel in subcutaneous vesicles. In contrast, subcutaneous vesicles are mostly characterized by structural proteins [29].

To get a wider picture of adipose tissue-secreted vesicles, we performed to the best of our knowledge, the first proteomic characterization of EVs secreted by brown adipose tissue from obese and lean rats, which were compared to those vesicles secreted by white adipose tissues of the same animals; hence, we characterized their composition according to the anatomical location of the tissue and their nutritional status [18]. Brown fat, despite having a different metabolic role than white, plays an important role in cell communication, as does white, through the secretion of various batokines or peptide and non-peptide molecules, such as microRNAs and lipids [96]. So far, recent studies have characterized the proteomic profile of murine brown adipose tissue secretome [97], and of the primary culture of brown adipocytes from the supraclavicular region [98], setting a good starting point to characterize brown adipose tissue batokines. In this regard, it is worth highlighting that the study by Deshmukh et al. showed that among those identified proteins in the secretome of brown adipose tissue, some had vesicular origin.

**Table 2 biomedicines-11-00422-t002:** Overview of the reviewed publications related to the characterization of the protein load of EVs from adipose tissue or at the circulating level; origin, species, and EVs-biomarkers function.

EVs Origin	Species	EVs Biomarkers
Adipocytes of the SGBS line [41]	Human	FABP4, adiponectin, TNFα, MIF, RBP4
Brown adipocytes under thermogenic stress [61]	Mouse	PDHE1-B
Adipocytes of the 3T3-F442A line [44]	Murine	Involved in fatty acid oxidation
Adipocytes of line 3T3-L1 [63]	Murine	lVEs: FABP4, annexin-2, endoplasmin, actin-4 sVEs: FAS, adiponectin
Healthy and pathological adipocytes of the line C3H10T1/2 [25]	Murine	TGFBI, mimecan, ceruloplasmin, caveolin-1, perilipin-1
Primary culture of adipocytes under obese and diabetic conditions [64]	Rat	Caveolin, aquoporin-7
Abdominal fat MSCs [62]	Pig	VEGF
Subjects undergoing surgery for aortic aneurysm (obese/overweight) [44]	Human	MCP-1, IL-6, MIF
Omental tissue from women with gestational diabetes [69]	Human	Glucose metabolism-related proteins
Subcutaneous and visceral adipose tissue from patients with obesity [29]	Human	TGFBI, caveolin-1, CD14, mimecan, thrombospondin-1, FABP-4, AHNAK, syntenin-1
Subcutaneous, visceral, and brown adipose tissue from animals with obesity and healthy [18]	Rat	UCP1, ATP citrate synthetase, vimentin, ceruloplasmin, FAS, FABP4
Inguinal fat [66]	Rat	NPM3, DAD1
Plasma from patients with metabolic diseases [71]	Human	Perilipin-1
Plasma from subjects with metabolic syndrome [82]	Human	Adiponectin, adipsin, chimerin, DDP4
Plasma from obese subjects before and after bariatric surgery [83]	Human	FABP4
Plasma from patients with obesity and healthy [29]	Human	TGFBI, mimecan, caveolin-1

Interestingly in our study, EVs from both obese and lean brown adipose tissues had a significant mitochondrial component (40–45%), unlike EVs released from white fat [18,96]. Thus, vesicles released from brown adipose tissue were characterized by proteins of the beta-oxidation pathway [96]. Relevantly, we have not only identified the brown adipose tissue thermogenesis protein, UCP1 (mitochondrial uncoupling protein of brown fat 1), for the first time in brown adipose tissue EVs, but we have also detected that this protein was elevated in brown EVs compared to those vesicles secreted from white (subcutaneous and visceral) [18]. Interestingly, EVs from the brown adipose tissue of obese animals contain a higher ratio of UCP1 compared to EVs from the same tissue in normal-weight/control animals [18]. This result seems contradictory to previous research, since, although the brown fat depot is more resistant to inflammation than the white one, secretion of proinflammatory cytokines is known to alter the thermogenic function of brown adipose tissue by reducing the expression levels of UCP1 and other markers of thermogenesis under obese conditions [99]. In addition, reduced activation of the brown depot, specifically reduced expression of UCP1, has been described in murine models of genetic obesity [100]. In contrast, some studies of diet-induced obesity (DIO) indicate that UCP1 levels remain stable or, as in the work of Alcalá M. et al., UCP1 protein levels are up-regulated in obese mice after a high fatty acid diet (HFD) for 20 weeks [101]. Therefore, we believe that EVs released from brown fat represent the adaptive state of this depot, which triggers an early adaptive thermogenic response induced by an HFD diet, reflected by an increase in UCP1 protein levels. On the other hand, this UCP1 protein has also been identified in visceral adipose tissue EVs from obese animals compared to subcutaneous EVs from normal-weight and obese animals [18]. Other studies confirm this finding, as a higher level of UCP1 gene expression has been observed in visceral depots of obese patients compared to subcutaneous depots of lean and obese patients [102].

On the other hand, other enzymes and proteins related to the insulin signaling pathway, and cytokines, identified in brown adipose tissue, secreted EVs, such as C3/C4, AOC3 (primary membrane amine oxidase), MIF1 (macrophage migration inhibitory factor), REEP5 (receptor enhancer protein 5 expression), PEPCK-C (phosphoenolpyruvate carboxykinase), TARG1 (regulator of GLUT4 trafficking 1), and Gyk (glycerol kinase), which were also identified in the secretome of murine and human brown adipose tissues [18]. Proteins of the same family as REEP5, such as REEP6, are known to regulate cold-induced thermogenesis, showing that a deficiency of this protein leads to an obesity-prone phenotype [103]. Furthermore, overexpression of PEPCK-C, a glyceroneogenesis-regulating enzyme in adipose tissue, has been reported to lead to a high susceptibility to insulin resistance and diet-induced obesity [104]. Additionally, in this pathological context, numerous reports have shown an increase in circulating AOC3 in diabetic conditions; thus, it has been described to be up-regulated in the adipose tissue of obese/diabetic rodents [105]. The TARG1 protein is also known to positively regulate GLUT4 trafficking and insulin sensitivity in adipocytes [106]. In addition, it has also been postulated that the Gyk protein, an important gene in the browning process of white adipocytes, stimulates UCP1 expression through the activation of the βAR-cAMP-CREB pathway and regulation of lipid metabolism [107]. Interestingly, the Gyk enzyme has been shown to have higher activity in brown adipose tissue than in white in rats and mice, paralleling the findings in brown adipose tissue EVs [108].

In addition, EVs from brown deposits of normal-weight animals contained up-regulated proteins such as ACLY (ATP citrate synthetase), Rab 14, ACC (acetyl-CoA carboxylase), and the protein 2-enoyl thioester reductase, compared to EVs from obese brown deposits [18]. ACLY is a protein involved in lipogenesis, and it is known that during the development of obesity there is a decrease in this protein [109]. On the other hand, the Rab 14 protein is involved in the transit of GLUT4, internalizing it from early endosomes to the Golgi complex [110]. Furthermore, ACC and 2-enoyl thioester reductase, anabolic de novo lipogenesis genes, are known to be activated under thermogenic conditions correlating with UCP1 expression [111]. Therefore, brown adipose tissue secretes EVs, which differ in their composition according to the metabolic state and dynamically reflect the changes produced in the cell of origin; thus, these EVs carry biomarkers of mitochondrial brown activity, oxidative stress, and inflammation during the development of obesity [18] (Figure 1).

## 3. Adipose Tissue Extracellular Vesicles’ Functional Role

The role of EVs, as implicated in the development of metabolic diseases, is still poorly understood [15]. Increasing evidence implicates EVs in obesity-associated metabolic dysregulation and, more specifically, in local and systemic inflammation related to liver and adipose tissues [112]. Thus, it has been proposed that EVs shed by adipose tissue are involved in the interaction between adipocytes and macrophages, and that they affect insulin signaling and expression of genes related to energy homeostasis control in muscle and liver cells, leading to metabolic diseases [15]. Since EV biogenesis is a dynamic process and depends on the pathophysiological state of the cell of origin [11], the functional role of EVs released by adipose tissue will differ depending on the molecular composition of these EVs (Figure 1 and Table 3).

Different functional assays have been performed in which EVs released by adipose tissue interact with metabolic target cells or tissues involved in the dysregulation of obesity disease. Thus, we have described that EVs from adipocytes of the C3H10T1/2 cell line with lipid hypertrophy and insulin resistance induce and stimulate the differentiation and lipid hypertrophy of neighboring healthy adipocytes [25]. This agrees with previous studies in which EVs isolated from adipose tissue internalize, and, in turn, induce adipogenesis in adipose tissue-derived stem cells (ADSCs) [56]. On the other side, EVs from lipid hypertrophied and insulin resistant adipocytes were also observed to promote insulin resistance in these same neighboring adipocytes, as they inhibited the phosphorylation of several proteins involved in the insulin signaling pathway [25]. Another study, by Kranendonk et al., has found that EVs released from the subcutaneous fatty tissue of some individuals inhibit Akt phosphorylation in HepG2 liver cells and in C2C12 myotubes [39]. Furthermore, others have reported that vesicles released from adipose deposits induce insulin resistance in macrophages [40]. The study by Ferrante et al. has shown that EVs released by adipocytes of adipose tissue from obese individuals transfer microRNAs that target genes involved in inflammatory and fibrotic signaling pathways, such as miR-141-3p, which is responsible for normal Akt protein phosphorylation upon insulin stimulation [113]. It has also been observed that decreased miR-22 expression in serum EVs correlates negatively with markers of adipogenesis and insulin sensitivity, such as PPAR at the hepatic level, promoting lower insulin sensitivity and, thus, insulin resistance [114]. In the same context, the study by Gesmundo et al. has revealed that EVs released from healthy 3T3-L1 adipocyte cultures increased survival and proliferation of the INS-1E pancreatic β cells and also the human pancreatic islets, thereby improving insulin sensitivity. However, EVs isolated from whole explants of obese human adipose tissue caused death and dysfunction of human β EndoC-βH3 cells [115]. Additionally, the study by Jayabalan et al. shows that EVs isolated from the omental adipose tissue of pregnant women with gestational diabetes increased the expression of genes associated with glycolysis and gluconeogenesis in human placental cells compared to EVs from pregnant women without this pathology [65]. Interestingly, it has been observed that EVs isolated from adipocytes are able to cross the blood–brain barrier and regulate POMC (arcuate pro-opiomelanocortin) expression through hypothalamic mTOR signaling in vivo and in vitro, thus, affecting whole body energy intake [43]. Therefore, different studies show that EVs released by adipocytes under obese conditions can decrease insulin sensitivity and, thus, promote the state of insulin resistance in different cells/tissues, such as the adipose tissue itself, pancreatic, muscle, and liver cells, among others.

Under the above, we have also found that EVs secreted by C3H10T1/2 adipocytes with lipid hypertrophy induced inflammation of non-inflamed healthy macrophages [25]. It is noteworthy that, in this study, we observed that to promote inflammation, a higher concentration of EVs from oleic acid hypertrophied adipocytes was needed compared to those shed by palmitic acid hypertrophy to produce the same deleterious effect [25]; thus, this suggests that oleic acid is less harmful in this context than palmitic acid as previously described [116]. Under this premise, other studies have shown that EVs derived from adipose tissue, especially those derived from visceral fat, have a proinflammatory effect on macrophages [41]. The study by Deng et al. has shown that EVs from adipocytes of obese mice increased the production of macrophage colony-stimulating factor (MCSF), IL-6, and TNF-α in primary murine macrophages [40]. In another report, EVs isolated from adipose tissue macrophages of obese mice caused impaired glucose tolerance and insulin sensitivity by tail vein injection in lean mice [92]. Interestingly, the study by Ogawa et al. observed that macrophages treated with EVs released from adipocytes expressed adipocyte-specific transcripts, such as adiponectin and resistin, and this was mainly due to the transfer of microRNAs [117].

On the other hand, it has also been observed that EVs secreted by macrophages from both mouse and human obese adipose tissue suppress insulin secretion and increase β-cell proliferation [54]. Interestingly, it was also shown that EVs from cultured Raw 264.7 macrophages containing miR-210 increased glucose uptake and mitochondrial activity in murine 3T3-L1 adipocytes in contrast to what happened when miR-210 was inhibited, which mitigated the insulin resistance effects promoted by these EVs [118]. Furthermore, in a study by De Silva et al., it was observed that primary adipocytes isolated from omental tissue, which was treated with EVs from lipopolysaccharide (LPS), activated the human macrophage cell line THP-1, showed no adipocyte differentiation and fat padding but induced changes in the gene expression of inflammatory pathways in adipocytes [119].

In the study by Zhao et al., they observed that small EVs (sEVs) secreted by mouse brown adipose tissue after exercise participate in exercise-associated cardio-protection through the delivery of miRNAs into cardiomyocytes, playing a cardioprotective role in the heart [120]. In this context, other investigators have observed that EVs released from perivascular adipose tissue containing miR-382-5p reduce foam cell formation in macrophages by up-regulating cholesterol transporters, in contrast to EVs released from SAT, which contain low levels of this miRNA [121]. Furthermore, Kariba et al. have observed that EVs isolated from brown adipose tissue containing mir-132-3p suppressed Srebf1 expression in a primary culture of murine hepatocytes, thereby attenuating the expression of lipogenesis at the hepatic level [58].

Regarding the association of obesity with a higher risk of developing cancer, several studies have shown that EVs isolated from adipocytes of human obese adipose tissue may be pro-oncogenic, promoting the proliferation of MCF7 tumor breast cells and other breast tumor cell lines within 24 h of treatment [122,123]. Additionally, research by Lin et al. shows that EVs isolated from mesenchymal stem cells from human adipose tissue isolated by liposuction promoted proliferation and migration of MCF7 cells [124]. Likewise, in vitro studies suggest that adipocyte-derived EVs can promote migration and proliferation of HCC hepatocarcinoma cells [57]. Other studies also observed that EVs released by adipocytes have a prooncogenic effect and promote the aggressiveness of cutaneous melanoma [44,125].

Finally, as brown adipose tissue EVs and their functional role is still an emerging field, it should be mentioned that the research by Zhou and collaborators shows that brown adipose tissue secreted exosomes that mitigated the metabolic syndrome in animals fed with a high fat diet [19]. They show how brown exosomes reduced body weight, lowered blood glucose, alleviated lipid accumulation, promoted oxygen consumption of cells, and restored abnormal cardiac functions accumulation in HFD mice independently of food intake.

**Table 3 biomedicines-11-00422-t003:** Summary of the reviewed publications related to the functionality of the EVs in the context of obesity.

EVs Origin	Function(s)	Target Cell/Tissue
Control and pathological C3H10T1/2 murine adipocytes [25]	Increase of adipose differentiation and insulin resistance	C3H10T1/2 adipocytes
Control and pathological C3H10T1/2 murine adipocytes [25]	Increase inflammation	Raw 264.7 macrophages
Inguinal adipose tissue [52]	Increase adipogenesis	Adipose tissue derived stem cells
Healthy 3T3-L1 adipocyte cultures [115]	Increase survival and proliferation	INS-1E pancreatic β cells
Human subcutaneous adipose tissue [44]	Promote insulin resistance	HepG2 liver cells C2C12 myotubes
Omental adipose tissue of pregnant women with gestational diabetes [69]	Increase the expression of genes associated with glycolysis and gluconeogenesis	Human placental cells
Mouse visceral adipose tissueadipocytes [54]	Regulate POMC (arcuate pro-opiomelanocortin) expression through hypothalamic mTOR signaling	Mouse, in vivo
Adipocytes of obese mice factor [40]	Increase the secretion of MCSF, IL-6 and TNF-α	Primary murine macrophages
Murine adipocytes culture [117]	Transfer specific microRNAs, such as adiponectin or resistin on the target cells	Murine macrophage cells
Raw 264.7 macrophages [118]	Increase glucose uptake and mitochondrial activity	Murine 3T3-L1 adipocytes
Human macrophage activated by lipopolysaccharide [119]	Induce changes in the gene expression of inflammatory pathways	Primary adipocytes isolated from omental tissue
Mouse brown adipose tissue after exercise [120]	Play a cardioprotective role	Murine heart cells
Brown adipose [58]	Attenuate the expression of lipogenesis genes	Primary culture of murine hepatocytes
Mesenchymal stem cell from human adipose tissue isolated by liposuction [124]	Promote proliferation and migration	MCF7 cells
Brown adipose tissue from mouse [19]	Mitigate the metabolic syndrome	Mouse, in vivo

## 4. Conclusions

Currently, extracellular vesicles are attracting exponential interest owing to their signaling role and as transporters of adipokines and biomarkers of the cell and tissue of origin, which dynamically reflets pathological deregulation; hence, they also provide new promising therapeutic options. Even though studies on the characterization and functional role of EVs released by adipose tissue took some time to develop, unlike other cellular systems, such as in the immune system or in diseases such as cancer, in recent years, the number of investigations regarding EVs released in the context of metabolic regulation has increased. Consequently, the potential role of EVs in adipose tissue communication at the local and peripheral/central levels has been demonstrated.

In this review, we tried to assemble all the research performed to date in relation to the characterization of molecular cargo and the function of adipose tissue EVs, focusing on their modification under metabolic stress and/or in the context of obesity and its comorbidities. Hence, we can conclude that adipose tissue can also secrete adipokines independently of the classical pathway by packaging those into extracellular vesicles. Furthermore, as these vesicles are sophisticated structures, they can target their contents to specific locations thanks to their membrane antigens that may be cell- or tissue-specific. Thus, adipose tissue can send signals both at the autocrine/paracrine and endocrine levels, interacting with distant tissues able to alter the gene expression of target cells due to the microRNAs transported inside. The research performed to date has shown that obesity is characterized by increased secretion of EVs both locally and at the circulating level. In addition, it can be concluded that obese adipocytes release a greater number of EVs whose charge and molecular composition vary according to their anatomical location. Therefore, in relation to the protein load, the visceral deposits secrete a greater number of different vesicular proteins including those related to obesity and its comorbidities, unlike the EVs released by the subcutaneous adipose tissue. Additionally, it must be highlighted that specific exosomal miRNAs liberated by adipose tissue may be implicated in different local and systemic deleterious aspects associated with obesity, such as glucose tolerance, insulin resistance, inflammation, cardiovascular and fatty liver disease, and even participating or fueling tumor growth. Future goals will need to consider the possibility of using the functional effects of adipose tissue-EVs for therapeutical purposes, which may represent important curative options to treat obesity.

## Figures and Tables

**Figure 1 biomedicines-11-00422-f001:**
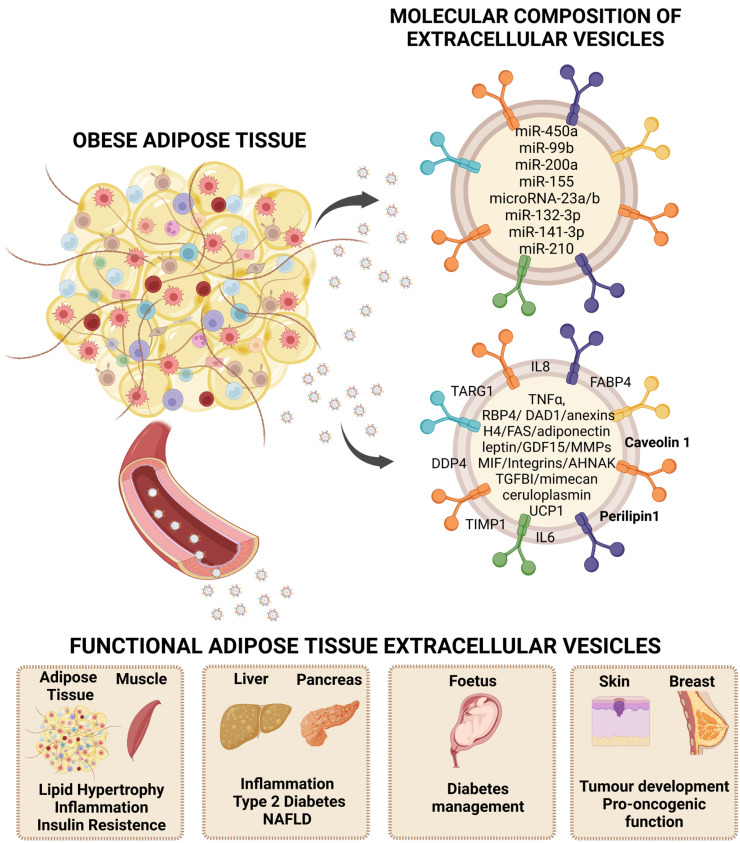
Extracellular vesicles (EVs) as carriers of adipokines and their role in obesity. Adipose tissue secretion of EVs as an alternative way of communication at the paracrine and endocrine levels that reflects characteristics of the metabolic status, thus, representing a rich source of disease biomarkers and exerting functional roles that participate in obesity-associated comorbidities. Representative bioactive cargo including microRNAs and protein content is shown. NAFLD: non-alcoholic fatty liver disease. Created by BioRender.

## Data Availability

Not applicable.

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
