# Peer review of "Extracellular Vesicles as Carriers of Adipokines and Their Role in Obesity"

_biomedicines, 2023, doi:10.3390/biomedicines11020422_

Round 1

Reviewer 1 Report

Major comments:

The article contains a lot of important information, however, it would gain significantly in the selection if the authors prepared figures and tables summarizing and comparing the presented dependencies in each of the chapters.

In addition, in my opinion, the summary should be enriched with the possibility of using the presented data in the future, in particular, drawing attention to the therapeutic goals of the listed diseases resulting from obesity.

Minor comments:

In line 36, please add the current WHO data.

In line 48, please add citation.

In line 55, please mention if there are already studies on the relationship between COVID-19 and the level of adipokines.

In line 177, plase sumarrize which differences are observed between obese and lean.

In line 181 diseases like... same in line 241

In line 379, plase indicate which animals. 

Author Response

Reviewer 1

Major comments:

  1. The article contains a lot of important information, however, it would gain significantly in the selection if the authors prepared figures and tables summarizing and comparing the presented dependencies in each of the chapters.

We are grateful to the reviewers for this comment and agree that it is necessary to collate the information for each section in a figure or table. Therefore, we have summarized this information in 3 tables, one for each section: Table 1 (EVs-miRNAS), Table 2 (EVs-proteins), and Table 3 (EVs-Functionality).

  1. In addition, in my opinion, the summary should be enriched with the possibility of using the presented data in the future, in particular, drawing attention to the therapeutic goals of the listed diseases resulting from obesity.

Please see that we have added this future remarks.

Minor comments:

  1. In line 36, please add the current WHO data.

We have added the required data. Please check line 36.

  1. In line 48, please add citation.

It has been corrected. Please check line 49.

  1. In line 55, please mention if there are already studies on the relationship between COVID-19 and the level of adipokines.

We appreciate the editor's suggestion and have added information on altered adipokine levels in patients with COVID-19 (reference 5).

  1. In line 177, please sumarrize which differences are observed between obese and lean.

We have summarized the differences between obese and lean people. Please confirm that it has been corrected on line 174.

  1. In line 181 diseases like... same in line 241.

This has been corrected. Please check line 184 and 244.

  1. In line 379, please indicate which animals. 

It has been corrected. Please check line 389.

Reviewer 2 Report

Dr. Pardo and colleagues presented an interesting review of the literature on adipose tissue-derived extracellular vesicles(EV)´ molecular cargo content, focusing on adipokine-related proteins in the context of obesity.

The group has experience in the proteome of adipose tissue-derived EVs, emphasizing their work in brown adipose tissue during obesity. They described significant differences between BAT´s secretome and the visceral AT. Consistently they also raised discussions on the differential content of AT-derived EVs from different depots (visceral and subcutaneous), which, in obesity, carry a greater number of obese- and comorbidities-associated molecules. They also presented additional data about the EV´s package, including miRNAs, lipids / free fat acids, and a vast panel of proteins, such as ECM, receptors, and adipokines. Those EVs released by adipose tissue (BAT or WAT/ visceral or subcutaneous) in obesity, by reaching the target cells, can affect gene expression, metabolism, and several cell functions, acting paracrine, autocrine, and endocrine altering body homeostasis.

Minor issues:

The manuscript is generally well structured and correctly based on the relevant literature. However, at some points in the text, data were quoted directly from other reviews used as references (especially in the first part of the manuscript).

At some points in the text (item 3), information is repeated, being unnecessary to be mentioned again. Ex: lines 589-592

The authors must check some of the information in the text based on other articles, which, in the way they are posed, are challenging to understand and may cause misinterpretation.

On Lines 269-270:  Please, review the information cited:

 ….miR-155 was found to be overexpressed in obese ATM exosomes and proved to SUPPRESS INSULIN EFFECT ON GLUCOSE PRODUCTION by downregulating its target, PPARγ mRNA [57].

Did those authors demonstrate the insulin effect on glucose production?

Please, check that reference 57 (Ying et al., Cell 2017) mentions that “miR-155KO animals are insulin sensitive and glucose tolerant compared to controls”.

On lines 457: When authors mention:  [82]. Furthermore, THIS protein… please identify the protein” (GLP1 or GIP?   Other?)

It would be fine if the authors could review the text in English grammar.

Author Response

Reviewer 2

Minor issues:

  1. The manuscript is generally well structured and correctly based on the relevant literature. However, at some points in the text, data were quoted directly from other reviews used as references (especially in the first part of the manuscript).

We appreciate this comment from the reviewers, and we have revised the article and reorganized the information collated from the different publications.

  1. At some points in the text (item 3), information is repeated, being unnecessary to be mentioned again. Ex: lines 589-592

We agree with the reviewers and have reorganized that part.

  1. The authors must check some of the information in the text based on other articles, which, in the way they are posed, are challenging to understand and may cause misinterpretation.

It has been checked and mended.

  1. On Lines 269-270: Please, review the information cited:  ….miR-155 was found to be overexpressed in obese ATM exosomes and proved to SUPPRESS INSULIN EFFECT ON GLUCOSE PRODUCTION by downregulating its target, PPARγ mRNA [57].Did those authors demonstrate the insulin effect on glucose production?

The authors of this article (Ying et al., Cell 2017) have performed glucose tests, insulin sensitivity, and have also confirmed the levels of GLUT4 and PPARγ by immunodetention techniques.We have added this information in the article.

  1. Please, check that reference 57 (Ying et al., Cell 2017) mentions that “miR-155KO animals are insulin sensitive and glucose tolerant compared to controls”.

We agree with the reviewers and have organized and added the suggested information in this part of the article.

  1. On lines 457: When authors mention: [82]. Furthermore, THIS protein… please identify the protein” (GLP1 or GIP?   Other?)

 It has been corrected. Please check line 467.

  1. It would be fine if the authors could review the text in English grammar.

This was revised.